# Incidence and Risk Factors of Postoperative Febrile Morbidity among Patients Undergoing Myomectomy

**DOI:** 10.3390/medicina59050990

**Published:** 2023-05-20

**Authors:** Korrakot Wattanasiri, Worashorn Lattiwongsakorn, Rung-aroon Sreshthaputra, Theera Tongsong

**Affiliations:** Department of Obstetrics and Gynecology, Faculty of Medicine, Chiang Mai University, Chiang Mai 50200, Thailand

**Keywords:** febrile morbidity, fibroid, fever, leiomyoma, myomectomy

## Abstract

*Background and Objectives:* To identify the incidence, causes, and independent predictors of postoperative febrile morbidity among patients undergoing myomectomy. *Material and methods:* Medical records of patients who had undergone myomectomy at Chiang Mai University Hospital between January 2017 and June 2022 were comprehensively reviewed. The clinical variables, including age, body mass index, previous surgery, leiomyoma size and number, the International Federation of Gynecology and Obstetrics (FIGO) fibroid type, preoperative and postoperative anemia, type of surgery, operative time, estimated blood loss, and intraoperative antiadhesive use, were analyzed as a predictive factor of postoperative febrile morbidity. *Results:* During the study period, 249 consecutive women were reviewed. The mean age was 35.6 years. The majority of women had FIGO fibroid type 3–5 (58.2%) and type 6–8 (34.2%). Febrile morbidity was noted in 88 women (35.34%). Of them, 17.39% had a urinary tract infection and 4.34% had a surgical site infection, whereas the causes in the majority of cases (78.26%) could not be identified. The significant independent risk factors for febrile morbidity were abdominal myomectomy (adjusted odds ratio: aOR, 6.34; 95% CI, 2.07–19.48), overweight women (aOR, 2.25; 95% CI, 1.18–4.28), operation time of more than 180 min (aOR, 3.37; 95% CI, 1.64–6.92), and postoperative anemia (aOR, 2.71; 95% CI, 1.30–5.63). *Conclusions:* Approximately one-third of women undergoing myomectomy experienced febrile morbidity. The cause could not be identified in most cases. The independent risk factors included abdominal myomectomy, overweight, prolonged operation time, and postoperative anemia. Of them, abdominal myomectomy was the most significant risk factor.

## 1. Introduction

Leiomyoma is the most common benign tumor in women, occurring in approximately 25% of reproductive women [1]. Leiomyoma can cause many problems, such as abnormal uterine bleeding, pressure symptoms, or infertility. Myomectomy is conservative surgical management, a typical option for reproductive women who suffer from leiomyoma and require preservation of their fertility. Various surgical approaches for myomectomy are used in practice, involving open (abdominal) laparotomy, laparoscopy, and hysteroscopy [2,3]. Both laparoscopic and abdominal myomectomies can dramatically improve quality of life of women with symptomatic leiomyomas [4]. We perform approximately 50 myomectomies per year in our tertiary care hospital. Fertility rates following myomectomy are 20–50% in the first year following surgery and up to 70% thereafter [5]. Nevertheless, either abdominal myomectomy or laparoscopic myomectomy is associated with significant complications. One of the most common complications is febrile morbidity, probably caused by infection, hematoma, or unknown cause [6]. The rates of febrile morbidity after myomectomy vary greatly from study to study, ranging from 12 to 67% [6,7,8,9,10]. The different rates are partly associated with use of different definitions of postoperative fever. Postoperative fever is significantly associated with increased duration of hospital stay and need for more interventions. Though, in most patients, postoperative fever is caused by a benign cause and resolves spontaneously, some patients need specific treatment, such as for urinary tract infection, deep vein thrombosis, pulmonary embolism, etc. Moreover, febrile morbidity may be the first sign of some serious conditions with worse prognosis, such as anastomotic leaks or bowel obstruction. Failure to diagnose the cause of fever or identify the severity of it can lead the patient into systemic inflammatory response syndrome (SIRS), sepsis, severe sepsis, or septic shock. This can lead to prolonged hospitalization and even increase the mortality rate. Therefore, risk factors and causes of febrile morbidity among these patients are clinically significant. Additionally, knowledge of risk factors for postoperative fever is expected to be clinically helpful. The cost-effectiveness of postoperative workup is closely related to its yield. For example, in a study on the value of postoperative evaluation [11], all patients with postoperative fever underwent a workup, resulting in only a 27% (19/71) yield of localized findings representing an infectious source of the fever, and 75% of them could have the correct diagnosis made based on clinical findings and confirmed by a single appropriate test.

Though febrile morbidity secondary to hysterectomy has been published several times, studies on that associated with myomectomy have been reported in a very limited number. Moreover, most previous studies included only abdominal myomectomy. To the best of our knowledge, no study has specifically focused on the comparison of febrile morbidity between abdominal and laparoscopic myomectomy. Therefore, we conducted this study to determine the incidence, causes, and independent predictors of postoperative febrile morbidity among patients undergoing both laparoscopic and abdominal myomectomy. The findings might be useful for developing a plan of treatment, counselling patients who are at high risk for febrile morbidity, and choosing pre- and postoperative antibiotics.

## 2. Patients and Methods

A retrospective analytical study was conducted at Maharaj Nakorn Chiang Mai University Hospital with ethical approval by the institute review board (the ethics committee of the faculty of medicine, Chiang Mai University, study research ID: 2565-08987). The study population was women undergoing myomectomy between January 2017 and June 2022. The patient identification numbers of all consecutive cases of women who underwent myomectomy during the study period were first taken from the hospital register of patient admissions and the operation room log book. Then, their medical records were comprehensively reviewed by the authors. On review, the following data were retrieved: (1) patients’ demographic data, including age at operation, body mass index (BMI), and parity; (2) clinical data, including medical comorbidity, previous surgery, fertility, type of operation, International Federation of Gynecology and Obstetrics (FIGO) fibroid classification type, number of leiomyomas, greatest dimension of tumors, clinical symptoms, main indication for myomectomy, use of preoperative gonadotropin releasing hormone (GnRH) agonist, preoperative anemia, and use of prophylactic antibiotic; (3) operative data: amount of intraoperative blood loss, operative time, blood transfusion, perioperative complications, and use of intraoperative antiadhesive agent; (4) postoperative data: postoperative anemia and postoperative complications, such as febrile morbidity and source of infection. Inclusion criteria included: (1) age of 18–55 years; (2) undergoing myomectomy by our reproductive medicine team; (3) postoperative pathological diagnosis of leiomyoma; and (4) providing written informed consent for the operation.

In our practice, abdominal or laparoscopic myomectomy was performed under sterile technique by the staff or fellows of the gynecologic reproductive medicine unit. A Foley catheter was inserted in the operating room and maintained for no more than 1 day postoperatively. Cefazolin 1 g was routinely given intravenously 30 min before the operation and was repeated every 6 h after the operation, for a total of 3 doses. A preoperative GnRH agonist was used in cases of symptomatic treatment for control of heavy menstrual bleeding and correction of anemia and for those in need of tumor size reduction during long waiting periods for operations of more than 12 weeks. GYNECARE INTERCEED^®^ absorbable adhesion barrier [Ethicon Johnson & Johnson surgical technologies] was offered as an option for the patients. Concerning surgical techniques, though the techniques of abdominal and laparoscopic surgeries were based on the surgeon’s discretion, all were performed by the same reproductive medicine team using the same standard techniques. Typically, the myometrial incision was closed using running, unlocked layers of 0 Vicryl suture (polyglactin) on a circle taper needle. Two-layer closure of myometrium was used for a larger defect (greater than 2 cm in depth). The uterine serosa was closed with a continuous, unlocked stitch using 1- or 2-0 Monocryl. The peritoneum was closed with 2-0 Vicryl, whereas the fascia was closed with 0 Vicryl.

Febrile morbidity was defined as a body temperature of 38.0 °C or higher, taken by mouth as a standard technique at least four times daily, that occurred in any two consecutive days of the first 10 days postoperative, excluding the first 24 h. A specific postoperative infection was diagnosed by clinical symptoms and signs and/or laboratory tests. Typically, the patients were discharged within 2–3 days. They were instructed by well-trained nurses to take a temperature at home using a standard oral thermometer 4 times a day. It was emphasized that once the fever was noted, the health care provider must be notified. The patients with febrile morbidity were admitted to the hospital and underwent a complete physical examination and basic laboratory tests, such as a complete blood count and urinary analysis, as well as a pelvic ultrasound examination and a specific workup depending on the clinical manifestations. For example, if pneumonia was suspected, a chest X-ray might be ordered; blood and urine might be cultured for sepsis in case of suspicion of a urinary tract infection; and duplex ultrasound might be ordered if a deep vein thrombus was suspected. The treatment depended on the causes; antibiotics were used for infectious causes, and symptomatic treatment with acetaminophen was often used as needed, especially in cases of unknown causes; typically, the oral dose was 650 mg every 4 to 6 h.

The statistical analysis was performed using the Statistical Package for the Social Sciences (SPSS) software, version 26.0 (IBM Corp., released 2019, IBM SPSS Statistics for Windows, version 26.0; IBM Corp., Armonk, NY, USA). Descriptive statistics were used for demographic baseline data. The chi-square or Fisher exact test were used to univariately identify factors related to the presence of febrile morbidity. Multivariate analysis using the logistic regression model was performed to find the independent risk factors. An odds ratio, with a 95% confidence interval that did not include unity was considered statistically significant. According to the primary outcome, with an estimation of febrile morbidity prevalence of about 31% [6], this study needed a sample size of at least 225 cases to gain a power of 90% at a 95% confidence interval.

## 3. Results

During the study period, 249 women meeting the inclusion criteria were included in the study. All were of Thai ethnicity and lived in the northern part of Thailand. The mean age was 35.6 ± 4.9 years and the mean body mass index (BMI) was 22.3 ± 3.9, which was within the normal range for Asian women, as presented in Table 1. Most of them had no history of medical comorbidity, anemia, or previous surgery. The majority of cases (94.4%) were nulliparous.

Two hundred and twenty-six (90.8%) of the women had a tumor size of greater than 4 cm at the widest diameter, and approximately two-thirds (65.8%) had a number of tumor masses of 1–3. The most common types of fibroids, according to FIGO classification, were types 3–5, accounting for 58.2%, followed by types 6–8 (34.2%). Note that 15.7% of the women received a monthly preoperative GnRH agonist for tumor size reduction during a long waiting period for the operation, and 37.5% of the women had preoperative anemia.

Two hundred and twenty-one (84.7%) women underwent abdominal myomectomy, and the remaining 38 (15.3%) women underwent laparoscopic myomectomy. Most cases had an operative time of less than 180 min (68.7%), estimated blood loss of less than 1500 mL (90.4%), and no need for a blood transfusion (77.9%). Antiadhesive agents were used in 44.2% of cases.

Febrile morbidity was identified in 88 of the 249 women, with an incidence of 35.34% (95% confidence interval: CI = 33.84–36.22). Of them, the causes of febrile morbidity were identified in only a minority of cases (8 cases; 9.1%), and included urinary tract infection (5 cases; 5.7%), abdominal wound infection (1 case; 1.1%), and phlebitis (2 cases; 23%), whereas causes in the remainder (80 women) were unexplained. The mean ± standard deviation (SD) postoperative days of fever occurrence was 2.8 ± 3.1 days. All cases of infectious causes were treated with antibiotics, whereas the cases with unexplained causes received symptomatic treatment with acetaminophen or expectant management, and all resolved spontaneously without specific treatment.

Univariate analysis was performed to compare means or percentages of various risk factors between the groups with and without febrile morbidity, including age, body mass index (BMI), type of operation, parity, medical comorbidity, previous surgery, tumor size/number and FIGO classification, preoperative use of GnRH agonists, prophylactic preoperative antibiotics, preoperative anemia, operative time, intraoperative blood transfusion, and estimated blood loss, as presented in Table 2. On univariate analysis, the group with febrile morbidity showed significantly higher rates of cases with overweight women, abdominal myomectomy, preoperative anemia, operative time of more than 180 min, blood loss of more than 1500 mL, and postoperative anemia (*p*-value of less than 0.05). On multivariate logistic regression analysis, the risk factors that were still found significant included overweight (adjusted odds ratio: aOR, 2.25; 95% CI, 1.18–4.28), abdominal myomectomy (aOR, 6.34; 95% CI, 2.07–19.48), postoperative anemia (aOR, 2.71; 95% CI, 1.30–5.63), and operative time of more than 180 min (aOR, 3.37; 95% CI, 1.64–6.92). The operative time in the laparoscopic group was significantly longer than that in the abdominal myomectomy group (238 ± 119 vs. 155 ± 53 min; *p* < 0.001). Likewise, the number of cases with prolonged operation (>180 min) was significantly higher in the laparoscopic group (58/211: 52.6% vs. 20/38: 27.5%; *p*: 0.002). Note that the type of operation seems to have the highest impact on the occurrence of postoperative febrile morbidity, with the highest adjusted odds ratio for abdominal myomectomy.

## 4. Discussion

Insights gained from this study are as follows: (1) Approximately one-third of women undergoing myomectomy had postoperative febrile morbidity. (2) The risk factors significantly associated with such morbidity included overweight, pre- and postoperative anemia, open myomectomy, prolonged operation, and excessive blood loss. However, pre-operative anemia and excessive blood loss were not significant factors upon multivariate analysis. (3) The most strongly associated risk factor was the type of surgical approach, abdominal myomectomy. (4) The causes of febrile morbidity could not be identified in most cases in which the fever spontaneously resolved with expectant management.

The causes of postoperative febrile morbidity mainly include the following three: surgical site infection, non-surgical site infection, and unexplained. Theoretically, myomectomy may have a much greater risk of having febrile morbidity than simple hysterectomy because myomectomy is more likely to have the formation of an intra-myometrial hematoma and/or the release of nonspecific febrile factors from the myoma during dissection. The difficulty in making a definite diagnosis of surgical site of infection of intra-myometrial hematoma may explain the unknown cause of fever in this study, which accounted for more than 90% of cases. In this study, the incidence of febrile morbidity after myomectomy was 35.34%, including unexplained causes (91%), non-surgical infections (8%) and surgical site infections (1%). These results are consistent with the existing literature [6,7,9,10]. Nevertheless, the risk was much lower with laparoscopic myomectomy; only 18.4% of cases were noted to have such morbidity in this study. According to this study, whereas postoperative febrile morbidity was common, most was unexplained and spontaneously resolved. Therefore, a workup to identify the cause should be strongly selective rather than a routine investigation.

Abdominal myomectomy was the most strongly significant risk factor for febrile morbidity in the multivariate analysis. The risk for febrile morbidity in women undergoing abdominal myomectomy is 6.34 times that of laparoscopic myomectomy, consistent with the findings of the meta-analysis, which determined that the risk of postoperative fever in patients with laparoscopic myomectomy was 50% lower than among those treated with open surgery (OR 0.44, 95% CI 0.26 to 0.77) [12]. The significant effects of abdominal myomectomy on febrile morbidity might be associated with more extensive manipulation, higher amount of blood loss, and more exposure of visceral abdominal structures. This piece of evidence should be provided to the patients during counseling on their choices of types of operation. Notably, abdominal myomectomy per se was a strong independent risk factor, even after adjustment for other factors, such as operation time. Although abdominal myomectomy had significantly shorter operation times, which decreased the risk of postoperative fever, it still had a significantly higher risk. Also note that the cases with prolonged operation times (>180 min) were relatively high in this study, mainly caused by a significantly higher percentage in the laparoscopic group.

We demonstrated that postoperative anemia was an independent risk factor for postoperative fever. To the best of our knowledge, this finding has never been documented in previous studies. It is possible that anemia could aggravate tissue hypoxia or compromise local immunologic defense mechanisms, leading to subtle infection. Our findings may alert care providers to pay more attention to postoperative hemoglobin and avoidance of anemia.

The strength of this study is that the findings can represent the events in actual practice instead of the ideal conditions of research settings. Accordingly, the findings are more suitable for counseling patients in real practice.

Limitations of this study include: (1) The retrospective nature might provide less reliable data, which were exclusively extracted from medical records that were not always perfectly recorded in service settings. Selection bias could have existed; for example, patients with complicated problems tended to be selected for abdominal myomectomy rather than laparoscopic surgery. However, this bias might only be minimal since, after controlling for confounders, the effect on febrile morbidity was still significant. (2) Though the sample size may be adequate for the primary objective, it is too small for subgroup analysis of some variables; for example, only two cases with no prophylactic antibiotics were included. (3) Our results must be interpreted with caution and may not be perfectly generalized because baseline characteristics might be different from study to study.

## 5. Conclusions

Approximately one-third of women undergoing myomectomy experienced febrile morbidity. Overweight women, abdominal myomectomy, an operative time of more than 180 min, and postoperative anemia were significant independent risk factors for such morbidity. The type of operation seems to have the highest impact on the occurrence of postoperative febrile morbidity.

## Figures and Tables

**Table 1 medicina-59-00990-t001:** Baseline characteristics of the study population.

Characteristics	Mean ± SD/n (%): n = 249
Age (mean ± SD)	35.6 ± 4.9
BMI (mean ± SD)	22.3 ± 3.9
Type of Operation Exploratory laparotomy myomectomy Laparoscopic myomectomy	211 (84.7%) 38 (15.3%)
Parity Nulliparous women Parous women	235 (94.4%) 14 (5.6%)
Body Mass Index (kg/m^2^) Underweight (<18.5) Normal (18.5–22.9) Overweight (23–24.9) Obese (>25)	31 (12.4%) 130 (52.2%) 39 (15.7%) 49 (19.7%)
Medical Comorbidity Yes No	55 (22.1%) 192 (77.1%)
Preoperative Anemia Yes No	94 (37.8%) 155 (62.2%)
Previous Surgery Yes No	72 (28.9%) 177 (71.1%)
Preoperative GnRH Agonist >Yes No	39 (15.7%) 210 (84.3%)
Prophylactic Preoperative Antibiotics Yes No	247 (99.2%) 2 (0.8%)
Preoperative Antibiotics Yes No	42 (16.9%) 207 (83.1%)
Tumor Size <4 cm >4 cm	23 (9.2%) 226 (90.8%)
Number of Tumors 1–3 4–5 >5	164 (65.8%) 38 (15.3%) 47 (18.9%)
FIGO Classification of Fibroid 0–2 3–5 6–8	19 (7.6%) 145 (58.2%) 85 (34.2%)
Estimated Blood Loss (mean ± SD)	601 ± 668
Estimated Blood Loss <1500 mL >1500 mL	225 (90.4%) 24 (9.6%)
Operative Time <180 min >180 min	171 (68.7%) 78 (31.3%)
Antiadhesive Agent Used Yes No	110 (44.2%) 139 (55.8%)
Intraoperative Blood Infusion >Yes No	55 (22.1%) 194 (77.9%)
Postoperative Anemia Yes No	99 (39.8%) 150 (60.2%)
Postoperative hospital stay (day): median (IQR)	3 (2–5)

**Table 2 medicina-59-00990-t002:** Risk factors of febrile morbidity based on univariate and multivariate analysis.

	Postoperative Febrile		Univariate	Multivariate	
	Febrile (n = 88)	Non-Febrile (n = 161)	*p*-Value	*p*-Value	OR (95% CI)
Age (mean ± SD)	35.0 ± 5.6	35.8 ± 4.5	0.213	0.184	1.04 (0.98–1.11)
Body Mass Index (kg/m^2^)	23.1 ± 4.4	21.9 ± 3.5	0.015	-	-
Body Mass Index (kg/m^2^)					
Overweight (>23)	41 (46.6%)	47 (53.4%)	0.006	0.014	2.25 (1.18–4.28)
Normal (≤23)	47 (29.2%)	114 (70.8%)			
Type of Operation					
Abdominal myomectomy	81 (38.4%)	130 (61.6%)	0.018	0.001	6.34 (2.07–19.48)
Laparoscopic myomectomy	7 (18.4%)	31 (81.6%)			
Parity					
Nulliparous	82 (34.9%)	153 (65.1%)	0.545	0.980	0.98 (0.27–3.57)
Parous	6 (42.9%)	8 (57.1%)			
Medical Comorbidity					
Yes	22 (40.0%)	33 (60.0%)	0.442	0.331	1.43 (0.69–2.96)
No	66 (34.4%)	126 (65.6%)			
Preoperative Anemia					
Yes	42 (44.7%)	52 (55.3%)	0.016	0.876	1.06 (0.54–2.07)
No	46 (29.7%)	109 (70.3%)			
Previous Surgery					
Yes	29 (40.3%)	43 (59.7%	0.299	0.234	1.53 (0.76–3.10)
No	59 (33.3%)	118 (73.3%)			
Preoperative GnRH Agonist					
No	77 (36.7%)	133 (63.3%)	0.310	0.299	1.60 (0.66–3.91)
Yes	11 (28.2%)	28 (71.8%)			
Prophylactic Antibiotics					
No	1 (50.0%)	1 (50.0%)	0.663	0.917	1.17 (0.06–22.71)
Yes	87 (35.2%	160 (64.8%)			
Tumor Size					
<4 cm	6 (26.1%)	17 (73.9%)	0.330	0.320	0.55 (0.17–1.78)
≥4 cm	82 (36.3%)	144 (63.7%)			
Number of Tumor Masses					
1–3	61 (37.2%)	103 (62.8%)	0.696	0.308	-
4–5	12 (31.6%)	26 (68.4%)		0.146	0.53 (0.23–1.25)
>5	15 (31.9%)	32 (68.1%)		0.647	0.78 (0.27–2.24)
FIGO Classification of Fibroid					
0–2	6 (37.5%)	12 (62.5%)	0.528	0.521	-
3–5	56 (37.8%)	92 (62.2%)		0.575	1.48 (0.38–5.79)
6–8	26 (30.6%)	59 (69.4%)		0.466	0.77 (0.38–1.56)
Estimated Blood Loss					
≥1500 mL	14 (58.3%)	10 (41.7%)	0.013	0.435	1.60 (0.49–5.16)
<1500 mL	74 (32.9%)	151 (67.1%)			
Operative Time					
≥180 min	39 (50.0%)	39 (50.0%)	0.001	0.001	3.37 (1.64–6.92)
<180 min	49 (28.7%)	122 (71.3%)			
Use of Antiadhesive Agent					
No	53 (38.1%)	86 (61.9%)	0.301	0.360	1.34 (0.71–2.52)
Yes	35 (31.8%)	75 (68.2%)			
Intraoperative Blood Infusion					
Yes	25 (45.5%)	30 (54.5%)	0.075	0.256	1.76 (0.67–4.63)
No	63 (32.5%)	131 (67.5%)			
Postoperative Anemia					
Yes	48 (48.5%)	51 (51.5%)	<0.001	0.008	2.71 (1.30–5.63)
No	40 (26.7%)	110 (68.3%)			

## Data Availability

The datasets analyzed during the current study are available from the corresponding author upon reasonable request.

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
