# Peer review of "Incidence and Risk Factors of Postoperative Febrile Morbidity among Patients Undergoing Myomectomy"

_medicina, 2023, doi:10.3390/medicina59050990_

Round 1

Reviewer 1 Report (Previous Reviewer 1)

Thank you for making many of the changes to the manuscript. It is significantly improved. 

Line 52: define "SIRS" before you use the acronym

Line 135: define "BMI" before using acronym

Same with "FIGO," "GnRH", "SD", and "CI"

Improved and appropriate. 

Author Response

Comments and Suggestions for Authors

Thank you for making many of the changes to the manuscript. It is significantly improved.

Line 52: define "SIRS" before you use the acronym

Response: In revised MS, the fullterm text is provided as highlighted.

Line 135: define "BMI" before using acronym

Response: In revised MS, the fullterm text is provided as highlighted.

Same with "FIGO," "GnRH", "SD", and "CI"

Response: The full texts of “FIGO”, “GnRH”, “SD”, and “CI” are provided as highlighted.

Reviewer 2 Report (Previous Reviewer 2)

The submitted manuscript raises some doubts: The incidence of major complications in the analyzed cohort is unusually high: a) In particular, the cut-off for normal and high blood loss was obviously set at 1500 ml! b) 24 patients had a blood loss of more than 1500 ml, c) 55 pats required a blood transfusion (these are the double pats with blood loss > 1.5 L, what is the explanation?), d) In 31% of the operations, the operation time was longer than 3 hours. Impact of the learning curve on e.g. OP duration is likely. All these data deviate dramatically from the data known from the literature and make the results hardly comparable with other reports. In addition, 211 pats underwent abdominal surgery, but only 38 laparoscopic approach: This distribution differs from today's approach to uterine myomectomy. In addition, the cohorts are not separately analyzed - they are hardly comparable, but this fact does notjustify to melt the both groups together).

In addition, it is understandable to create a variable (overweight/ obesity) that is not supported by evidence. With a mean BMI of 22.3 SD 3.9 kg, the overweight rate was marginal. In fact, 19.7% of the entire cohort had a BMI over 25! The WHO defines overweight and obesity as follows: overweight is a BMI greater than or equal to 25; and obesity is a BMI greater than or equal to 30. The authors use the cutoff of BMI 23 for overweight and BMI 25 for obesity - without justification. This results in a "significant" variable, which is not true, but then used for the analysis.

Author Response

Reviewer 2 (response as highlighted in red)

Comments and Suggestions for Authors

The submitted manuscript raises some doubts: The incidence of major complications in the analyzed cohort is unusually high: a) In particular, the cut-off for normal and high blood loss was obviously set at 1500 ml! b) 24 patients had a blood loss of more than 1500 ml, c) 55 pats required a blood transfusion (these are the double pats with blood loss > 1.5 L, what is the explanation?), d) In 31% of the operations, the operation time was longer than 3 hours. Impact of the learning curve on e.g. OP duration is likely. All these data deviate dramatically from the data known from the literature and make the results hardly comparable with other reports.

Response: Our data is based on comprehensive review of medical record, operative note, anesthetic notes etc and representing actual practice performed by staff members of reproductive team, not ideal conditions of research settings. Our results might be probably different from some other studies, but they are reflective of the facts in actual practice. We make a request to accept our evidence. Though the prevalence of some morbidities seem to be relatively high or different from some other studies, they do not compromise the reliability of analysis of the association of the risk factors and febrile morbidity.

In addition, 211 pats underwent abdominal surgery, but only 38 laparoscopic approach: This distribution differs from today's approach to uterine myomectomy

Response: In our settings and many hospitals, the prevalence of adominal approaches are still higher than laparoscopic surygery, probabaly due to higher cost. So this is not surprised in many developing countries. In comparison, it is not necessary to have the balanced number of cases (15%; 38 cases) is large enough proportion for comparison and adjustment in this multivariate analysis.

In addition, the cohorts are not separately analyzed - they are hardly comparable, but this fact does not justify to melt the both groups together).

Response: As mentioned in previous response, in fact, they were separated analyzed, in terms of comparing fever rate using both univariate and multivariate analysis but we did not compare all aspects of the two approaches since this study focuses on febrile morbidity and was aimed to identify potential risk factors of febrile morbidity. Route is considered as one of strong risk factor of fever. Actually, while we focus on fever, route was taken in considerations as two approaches.

While leiomyoma is a patient’s problem, we have two choices of route approach and we have to offer these two options to the patients. On counseling, we have to provide advantages and disadvantages of the two approaches. Therefore, we also have to provide information of febrile morbidity in this two approaches. Since the main objective of this study is to identify the risk factor for febrile morbidy, it is reasonable to compare this two approaches as well. Please accept our rationale. The comparison is statistically appropriate. Actually, we do not melt both groups together but multivariate analysis automatically separate each parameter for adjustment of their interaction and identifying independency.

Our result (Table 2) indicates that abdominal myomectomy is a strong independent risk factor of fever when compared with laparoscopic myomectomy. This is new insight rarely described in other studies. Thus, this comparison provides important information which is helpful in patient’s counseling. Febrile morbidity of the two approaches should also be taken into considerations. (We comment this point in “Discussion” as highlighted in red”.

In addition, it is understandable to create a variable (overweight/ obesity) that is not supported by evidence. With a mean BMI of 22.3 SD 3.9 kg, the overweight rate was marginal. In fact, 19.7% of the entire cohort had a BMI over 25! The WHO defines overweight and obesity as follows: overweight is a BMI greater than or equal to 25; and obesity is a BMI greater than or equal to 30. The authors use the cutoff of BMI 23 for overweight and BMI 25 for obesity - without justification. This results in a "significant" variable, which is not true, but then used for the analysis.

Response: Because Asian population has a smaller body size than the western. The definition of obesity based on standard WHO definition is not appropriate for our population. We use the cutoff of BMI 23 for overweight and BMI 25 for obesity; based on WHO critera (for Asia Pacific population or The Regional Office for Western Pacific Region of WHO, the International Association for the Study of Obesity and the International Obesity Task Force also proposed a separate classification for obesity in Asia in 2000.) (World Health Organization. The Asia Pacific Perspective- Redefining Obesity and Its treatment 2000 Geneva WHO).  Several studies demonstrate that WHO: BMI for Asian population is better than standard WHO classification; for examples:

  • Verma M, Rajput M, Kishore K, Kathirvel S. Asian BMI criteria are better than WHO criteria in predicting Hypertension: A cross-sectional study from rural India. J Family Med Prim Care. 2019 Jun;8(6):2095-2100. Pan WH, Yeh WT.
  • How to define obesity? Evidence-based multiple action points for public awareness, screening, and treatment: an extension of Asian-Pacific recommendations. Asia Pac J Clin Nutr. 2008;17(3):370-4.

Reviewer 3 Report (New Reviewer)

The article entitled “Incidence and Risk factors of Postoperative Febrile Morbidity among Patients 2 Undergoing Myomectomy” is informative for clinicians as well as researchers. The authors of the study aimed to aimed to determine the incidence, causes and independent predictors for 66 postoperative febrile morbidity among patients undergoing both laparoscopic and abdominal myomectomy. The findings might be useful for a plan of treatment and counselling patients, who are at high risk for febrile morbidity, and choice of pre and postoperative antibiotics. A febrile morbidity affected almost one-third of the women having myomectomy surgery. In most instances, the cause could not be determined. Abdominal myomectomy, being overweight, lengthy operations, and postoperative anemia were the independent risk factors. Abdominal myomectomy was the most potent and important risk factor of all of them.

This study is well-designed, and manuscript is good structured and well-written.

The Introduction section, as well as methodology, is succinct and provide sufficient information about previously published research regarding the current theme, and applied methodological tools in the present pilot study. Text of results and Tables were presented clearly.

Having in mind obtained results, and medical significance of the topic, study itself is of scientific merit.

However, some minor issues should be ameliorated: English language and typographic errors should be corrected.

Abbreviations should be explained at first appearance in the text (for example SIRS).

Please in the Methods, enlist the inclusion criteria and replace written with written informed consent.

The references are uniformly presented.

English language and typographic errors should be corrected.

Author Response

Comments and Suggestions for Authors

The article entitled “Incidence and Risk factors of Postoperative Febrile Morbidity among Patients 2 Undergoing Myomectomy” is informative for clinicians as well as researchers. The authors of the study aimed to aimed to determine the incidence, causes and independent predictors for 66 postoperative febrile morbidity among patients undergoing both laparoscopic and abdominal myomectomy. The findings might be useful for a plan of treatment and counselling patients, who are at high risk for febrile morbidity, and choice of pre and postoperative antibiotics. A febrile morbidity affected almost one-third of the women having myomectomy surgery. In most instances, the cause could not be determined. Abdominal myomectomy, being overweight, lengthy operations, and postoperative anemia were the independent risk factors. Abdominal myomectomy was the most potent and important risk factor of all of them.

Response: Thank you for the comment.

This study is well-designed, and manuscript is good structured and well-written.

Response: Thank you for the comment.

The Introduction section, as well as methodology, is succinct and provide sufficient information about previously published research regarding the current theme, and applied methodological tools in the present pilot study. Text of results and Tables were presented clearly.

Response: Thank you for the comment.

Having in mind obtained results, and medical significance of the topic, study itself is of scientific merit.

Response: Thank you for the comment.

However, some minor issues should be ameliorated: English language and typographic errors should be corrected.

Response: English (typographic errors) has been reviewed and corrected.

Abbreviations should be explained at first appearance in the text (for example SIRS).

Response: In revised MS, all abbreviations are elaborated at the first presence, as suggested, as highlighted.

Please in the Methods, enlist the inclusion criteria and replace written with written informed consent.

Response: In revised MS, inclusion criteria are listed at the end of the first paragraph of “Methods” section.

The references are uniformly presented.

Response: Thank you for the comment.

Comments on the Quality of English Language

English language and typographic errors should be correcte

Response: English has been checked and corrected.

Round 2

Reviewer 2 Report (Previous Reviewer 2)

The authors did not implement any relevant changes to the manuscript.  Statements such as "Please accept our rationale" or "We make a request to accept our evidence" can be considered as insufficient. 

This manuscript is a resubmission of an earlier submission. The following is a list of the peer review reports and author responses from that submission.

Round 1

Reviewer 1 Report

Thank you for your manuscript on this important topic. 

Suggestions: 

1. Quality of English language, specifically grammar and syxtax, is poor. For example: line 38 - 39 should read "We perform approximately 50 myomectomies per year in our tertiary care hospital. Fertility rates following myomectomy is 20 - 50% in the first year following surgery, and up to 70% thereafter." Additional examples: the authors confuse singular and plural tense many times: line 49 should be "several times" not "time." These issues continue throughout the entire manuscript. I strongly recommend that the authors consider English language editing services with a native speaker to review and edit the grammar and syntax issues. 

2. Table 1: unreadable. Remove all the bullets as they are not needed in a table. Wrong use of parenthesis throughout the table which is distracting and erroneous. 

3. Results: please clarify the first paragraph regarding Asian people. The authors did not present race and ethnicity data at all, which is a limitation because the readers must assume that all patients are Asian given the location of the study. I would suggest adding race and ethnicity data to Table 1, or commenting somewhere in the Results that all patients are Asian. 

4. Methodology: what is the length of stay after the surgeries? This should be included in Table 1. There is no comment on which surgeries were outpatient and which patients remained in the hospital. How was the temperature checked and documented? Was this checked by the patients four times per day at home, and submitted to the clinic to be recorded? Was there a standard thermometer used by all patients for the purposes of the study? Were all patients who met criteria for fever seen and evaluated in the office for a workup to identify the cause? There needs to be a stronger description of the methods regarding how fever was diagnosed. 

I would like to see a description of the patients with febrile morbidity: including the average temperature elevation, when it occurred (number of postoperative days), workup involved, and treatment (were antibiotics prescribed? did the fever resolve spontaneously?)

5. In the Introduction, the authors spend time discussing the FIGO grading of fibroids. This is not relevant to the topic of fever, and they don't explain some of the reasons why fever can occur after myomectomy or hysterectomy. I would recommend re-writing the introduction to review the literature about the association of these surgeries with fever, instead of a generic overview of fibroid types. 

6. The authors mention that formation of a postoperative hematoma is the main reason why post-myomectomy fever occurs. However they do not describe whether this occurred in their patients: if there is suspicion of hematoma, ultrasound should be performed to work up these patients and see if this is occurring. 

7. In the Discussion, the authors suggest that longer duration of surgery is a potential reason why abdominal myomectomy is associated with increased risk of fever, compared to laparoscopic. I beg to differ: abdominal myomectomy is often associated with decreased operative time compared to MIS. If this is different in the authors' institution, they should back this statement up by presenting the time under anesthesia in their cohort. This was not included in the Results so it is not appropriate for them to include this as a discussion point. If possible, please go back and add this data. 

At this point, the paper needs significant revisions. I recommend going back to the original data and including more information. I also think the Methods need to be strengthened. There is merit in the topic, but it cannot be published in the current form. 

Reviewer 2 Report

The submitted work contains several methodological flaws and the reporting of the results further contributes to my critical opinion: 1) Since the open and laparoscopic approaches differ dramatically in terms of perioperative comorbidity, it is not understandable why the two approaches are not separated were analyzed. The two groups are not balanced and the results are likely to be biased given that only 15% of surgeries were performed laparoscopically. The results presented in Table 2 confirm these doubts as the approach (open vs. MIS) was identified as a significant variable. 2) The results are difficult to generalize as the perioperative data are alarming and clearly opposite to the literature data regarding laparoscopic myomectomies (compare please with Watrowski et al 2017, PMID: 28446936). a) Why did 22% of the patients required a blood transfusion? Which criteria were used? b) The mean estimated blood loss was > 600 ml, but only 40% of the patients had postoperative anemia. How can it be explained? Has the distribution of results (including possible confounders) been accounted for by statistical analysis? c) In 31% of the operations, the operating time was more than 3 hours. This fact should be better addressed. 3) Critical surgical steps are missing from the description of the technique used, which in turn can lead to misleasding results, e.g.: a) What suture material was used? Was it the same for open and laparoscopic surgeries? Was the technique for suturing the myometrium consistent across all surgeries (single layer, double or more layers, or at surgeon's discretion)? b) What does it mean that Interceed has been used in "patients who can afford"? 4) The perioperative management is controversial: a) Leaving the Foley in place for 24 hours could contribute to the morbidity: what was the reason for next-day catheterization? b) The low dose of prophylactic antibiotics (1 g cefazolin) were followed by a controversially long continuation (3 days). On the other hand, 17% of patients received no antibiotic prophylaxis at all. 5) The two tables contain formatting errors that make them difficult to read (see position of brackets, etc.). 6) The number and selection of references (9 references) are not suitable for serious work.